# How Can Bullying Victimisation Lead to Lower Academic Achievement? A Systematic Review and Meta-Analysis of the Mediating Role of Cognitive-Motivational Factors

**DOI:** 10.3390/ijerph18052209

**Published:** 2021-02-24

**Authors:** Muthanna Samara, Bruna Da Silva Nascimento, Aiman El-Asam, Sara Hammuda, Nabil Khattab

**Affiliations:** 1Department of Psychology, Kingston University London, Penrhyn Road, Kingston upon Thames KT1 2EE, UK; A.Elasam@kingston.ac.uk (A.E.-A.); sarahammuda@hotmail.com (S.H.); 2Department of Psychology, Brunel University London, London UB8 3PH, UK; bruna.dasilvanascimento@brunel.ac.uk; 3Department of Sociology, Doha Institute, Doha, Zone 70, Qatar; nabil.khattab@dohainstitute.edu.qa

**Keywords:** bullying, victimisation, bully/victims, academic achievement, motivation, mediation, meta-analysis, cognitive-motivational, academic engagement, self-esteem, self-efficacy

## Abstract

Bullying involvement may have an adverse effect on children’s educational outcomes, particularly academic achievement. However, the underlying mechanisms and factors behind this association are not well-understood. Previous meta-analyses have not investigated mediation factors between bullying and academic achievement. This meta-analysis examines the mediation effect of cognitive-motivational factors on the relationship between peer victimization and academic achievement. A systematic search was performed using specific search terms and search engines to identify relevant studies that were selected according to specific criteria resulting in 11 studies encompassing a sample total of 257,247 children (10 years and younger) and adolescents (11 years and older) (48–59% female). Some studies were longitudinal and some cross sectional and the assessment for each factor was performed by various methods (self, peer, teacher, school and mixed reports). Children involved in bullying behaviour were less likely to be academically engaged (k = 4) (OR = 0.571, 95% CI [0.43, 0.77], *p* = 0.000), to be less motivated (k = 7) (OR = 0.82, 95% CI [0.69, 0.97], *p* = 0.021), to have lower self-esteem (k = 1) (OR = 0.12, 95% CI [0.07, 0.20], *p* = 0.000) and lower academic achievement (k = 14) (OR = 0.62, 95% CI [0.49, 0.79], *p* = 0.000). Bullying involvement was also significantly related to overall cognitive-motivational factors (k = 17, OR = 0.67, 95% CI [0.59, 0.76], *p* = 0.000). Cognitive-motivational factors, taken together, mediated the association between bullying victimisation and academic achievement (k = 8, OR = 0.74, 95% CI (0.72, 0.77), *p* = 0.000). Bullying victimisation was negatively related to cognitive-motivational factors, which, in turn, was associated with poorer academic achievement. These findings were moderated by the design of the studies, assessment methods for the bullying reports, mediators and outcomes, country, age of children in the sample and/or types of bullying. The findings are of relevance for practitioners, parents, and schools, and can be used to guide bullying interventions. Interventions should focus on improving internal and external motivational factors including components of positive reinforcement, encouragement, and programs for enhancing academic engagement and achievement amongst children and adolescents.

## 1. Introduction

Involvement in bullying has been associated with short and long-term negative consequences including physical health issues and behavioural and emotional problems [1,2,3]. Such consequences also vary according to the role played in the bullying experience and whether the child is a victim, a perpetrator, or both (bully–victim). Externalising problems such as hyperactivity and conduct disorder have been typically reported among bullies, whereas internalising problems such as anxiety and mood disorders have been mainly observed among victims (e.g., [4,5]). In turn, bully–victims experience a more severe combination of internalising and externalising problems in comparison to victims or bullies only [1,6,7].

Bullying is prevalent in the school context and as such, on top of its consequences for mental health, bullying involvement has also been found to have a negative impact on children’s educational outcomes, particularly academic performance [8]. For example, a study in Norway demonstrated that bullying involvement in adolescents was associated with lower academic grades at an individual and school level [9]. In the United States, these findings were confirmed in a nationally representative sample of 7,304 students, after controlling for poverty, school size, and personal victimisation [10]. In a meta-analytical review of 33 studies, Nakamoto and Schwartz [8] confirmed the negative association between bullying victimisation and academic performance. Liu et al. [11] in a longitudinal study found that being bullied in 3rd grade predicted poor academic outcomes in 5th grade. In another longitudinal study, Juvonen et al. [12] found that for each 1-point (out of 4-point scale) increase in self-perceived victimisation, students’ Grade Point Average (GPA) decreased by 0.3-grade points. Wang et al. [13] also found that for every 1-point (out of 5-point scale) increase in peer victimisation, students’ GPA decreased by 0.44 units. Similar findings were noted by Van der Werf [14] who studied the effect of bullying on academic achievement. It was found that a 1 standard deviation (SD) increase in school bullying incidents resulted in a 0.55 SD standardised test score decrease in the short-term and 0.4 SD decrease in the long-term (two years) for affected students.

As well as being repeatedly associated with poor academic achievement (e.g., [8,15,16]), bullying victimisation has been associated with low self-esteem [17,18,19,20], low educational motivation [21], reduced academic self-concept (reading and mathematics) and lower commitment to study, and higher extrinsic motivation and test anxiety rates [22]. Some studies also found a negative association between peer victimisation and academic self-efficacy [23] and self-concept [24]. In addition, children who are victimised by their peers tend to have negative attitudes toward school [25], negative perceptions of school climate [13,26], and difficulties concentrating on school work [27].

Although the negative association between bullying and academic performance has been well documented, the underlying mechanism for this association is yet to be fully understood. Among the various mechanisms that may link bullying victimisation with academic achievement, cognitive–motivational variables such as academic motivation and aspirations have been recognised as important, but as yet under-researched, domains. For the purpose of this study, academic motivation is defined as the student’s interest and desire to engage in their school and learning activities [28], whilst academic aspirations are a student’s educational goals and choices [29]. In fact, students with high academic motivations and aspirations tend to succeed academically [30]. However, bullying victimisation has been found to reduce students’ motivations and aspirations [21,31]. As such, motivation and aspirations offer a plausible mediational path between bullying victimisation and academic achievement. Thus, this study aims to explore and review the literature that has investigated the indirect effect of bullying victimisation on academic achievement through cognitive-motivational variables, particularly academic motivation and aspirations.

### 1.1. Bullying Victimisation, Cognitive-Motivational Factors, and Academic Achievement

The expectancy-value theory [32,33] and the achievement goal theory [34,35] propose that individuals are more likely to engage in a particular task when such a task has some value to them and when they believe they are likely to do well. From this perspective, cognitive–motivational factors such as academic motivation and aspirations play an important role in explaining academic achievement. Consistent with this view, previous studies have demonstrated that students with higher motivation and higher aspirations are more likely to succeed academically than those with low motivation and low aspirations (e.g., [30,36,37]). Given that motivation and aspirations are important mechanisms for academic success, understanding how these factors can buffer against the damaging implications of bullying victimisation for academic achievement would be informative for the development of intervention programs.

The Self-Determination Theory (SDT) [38] offers a useful framework to understand the association between bullying victimisation and cognitive–motivational factors such as motivation and aspirations in the educational context. SDT postulates that relatedness, autonomy, and competence are three important factors to maintain positive well-being. Relatedness refers to the need for being connected to others [39], autonomy refers to the need for self-endorsement of an individual’s behaviour [40], whereas competence refers to the need for achieving attained goals [38]. From this perspective, in order to feel motivated and achieve their highest academic potential, all these three needs must be supported in students. Negative school conditions such as peer rejection, social exclusion, and bullying may undermine these needs [41,42,43].

In fact, bullying victimisation has been found to negatively influence students’ school relatedness, such that bullied students tend to feel less connected to their school and, in turn, tend to achieve poorly academically [43]. On top of this, students who have suffered bullying victimisation present lower academic motivation, reduced perceived academic competence [21] and lower educational aspirations [31] in comparison to their non-bullied peers. These consequences may also be long-lasting. For example, Goodboy et al. [44] found that students who were bullied in high school presented a low level of self-determined motivation, high levels of amotivation, and emotional, social, and institutional problems in their first semesters at university, which is likely to affect their academic achievement. Studies exploring the mechanism behind the association between bullying and academic achievement have found that bullying victimisation leads to higher psychological distress, which in turn reduces student engagement leading to lower academic achievement [45]. Fan and Dempsey [46], while controlling for gender and socioeconomic status, found that students that are victimised by their peers report lower academic motivation and self-efficacy, which results in lower academic achievement. Taken together these findings suggest that bullying victimisation reduces academic achievement by decreasing students’ motivation and aspirations.

### 1.2. The Present Study

Meta-analysis studies have demonstrated a negative association between bullying and academic achievement [8]; however, these studies fail to identify the potential underlying mechanisms for this association. In order to design effective intervention strategies to minimise the impact of bullying on academic achievement, there is a need for studies that investigate the mediators and mechanisms of this relationship. The factors linking bullying with academic achievement have only been tested empirically to a limited extent (e.g., [45,46]). However, an explanation for a phenomenon cannot emerge from the findings of a single study. Therefore, the main aim of this meta-analysis is to address this gap and identify and quantify the extent to which cognitive–motivational factors such as motivation and aspirations mediate the association between bullying involvement and academic achievement. In addition, although the academic achievement of bullies and bully–victims are also generally lower than that of those uninvolved in bullying, meta-analytical studies have mainly focused on bullying victimisation (e.g., [8]). Thus, this meta-analysis also aims to identify whether bullying subgroips and type of bullying involvement plays a role in the association between bullying, motivation and aspirations, and academic achievement.

Therefore, the current study will focus on addressing the gaps in the literature regarding the indirect effect of bullying involvement of all types on academic achievement through motivation and aspiration factors in children and adolescents. We hypothesise that the relationship between bullying victimisation and lower academic achievement is mediated by cognitive-motivational factors.

## 2. Materials and Methods

The meta-analysis followed PRISMA guidelines [47] (Appendix A).

### 2.1. Information Sources and Database Search

A literature search of all studies on bullying and academic motivation or aspirations, and bullying and academic achievement, published between January 2000 and January 2020, was undertaken. The following databases were selected as they incorporate pertinent disciplines. These include CINAHL (Current Index to Nursing and Allied Health Literature), Education Abstract, Education Research Complete, ERIC (Education Resources Information Center), PsychInfo, PubMed, and Web of Science. To identify all published and unpublished studies empirically analysing school bullying, academic achievement, and motivation and aspiration factors, we conducted systematic searchers by combining three different sets of keywords. The first set of keywords comprised terms describing “education” (i.e., education* OR academic* OR school), while the second set of keywords comprised terms describing achievement (i.e., achievement OR performance OR attainment OR success* OR motivation OR aspiration*), and the third group of keywords described “bullying” (i.e., bully* OR victim* OR bullied*). Studies were then selected based on specific inclusion and exclusion (see below). Both published and unpublished articles were selected and then further coded according to the variables examined by each study. More specifically, the studies were divided into: (1) studies that looked at mediation factors between bullying and academic achievement; (2) studies that looked at the association between bullying and academic achievement; and (3) studies that looked at the association between bullying and motivation or aspiration. Some studies belonged to more than one group. The final included mediation studies are the ones for which we could calculate the mediation.

### 2.2. Eligibility Criteria and Study Selection

#### 2.2.1. Inclusion Criteria

The inclusion criteria required that the study examine the link between bullying involvement, academic motivation or aspiration, and academic achievement in the same study. Second, the methodology had to be of a quantitative nature. Third, studies relating to traditional bullying (i.e., face-to-face bullying), including relational (i.e., purposeful damage and manipulation of peer relationships leading to social exclusion, spreading rumours) and/or direct bullying (i.e., physical such as hitting and pushing, and verbal such as making fun or insulting someone), and cyberbullying (i.e., bullying through digital electronic communication tools) were all included. Studies that referred to specific forms of bullying such as bullying focused on sexual orientation, where sexuality or gender are used against another person, were also included. Although the main aim of the study is to look at the mediation effect, those studies that did not necessarily explore mediation factors for the relationship between bullying and academic achievement, but that looked at either academic achievement or motivation or aspirations separately, were also retained. Fourth, the measures of bullying relationships, outcomes and mediators had to have been conducted through observational studies and/or various reporting methods (self, teacher, peer or school). Fifth, sufficient statistical information needed to be available in the study or provided by the authors for effect size calculation (e.g., means and standard deviations, odds ratio with 95% Confidence Interval, correlations, event rates and sample size, etc.). Sixth, participants needed to be children or adolescents (under 18 years of age). Lastly, articles in English, Portuguese, and Spanish were included.

#### 2.2.2. Exclusion Criteria

Studies that were qualitative, retrospective, intervention-based, meta-analyses or exclusively examined a clinical population were not included. Reference lists from meta-analyses studies were examined in order to ensure all relevant studies had been included.

### 2.3. Coding

There were two independent coders that categorised variables as relevant and any disparities were discussed and duly revised. Based on the search result, the studies were allocated to three categories: mediation studies, studies on academic achievement, and studies on cognitive-motivational factors. No studies that looked at the association between bullying and aspirations and academic achievement were found.

### 2.4. Coding of Study Characteristics and Moderators

The percentage by gender and age range of the participants were extracted from each study. One study [48] provided the school grade of the children instead of the age range and this was converted into age range according to the school system in the respective country that the study was performed in. The age was then categorised into childhood (5–10 years of age) and adolescence (11–18 years of age) or mix of both. The age range of the study that reported the grade will not be affected even if some students have repeated one year or more as this study was put in the adolescence group and repeating years would still have put these students in that category. The age category of the participants; the assessment method (child-report, peer-report, peer nomination, teacher-report, school-report, or mixed) of the predictor (bullying), the outcome (academic achievement) and the mediator (cognitive-motivational); the type of bullying (traditional, relational, general and cyber bullying, or mixed); bullying subgroups (bullies, victims, and bully/victims); the country in which the study was conducted; and the design of the studies (cross sectional vs. longitudinal) were all included in the meta-analysis as potential moderators.

Comprehensive Meta-Analysis (CMA) [49] was used to perform the analysis. Some articles did not report some of the essential data for the analysis of the indirect effect of bullying on academic achievement [15,50,51,52]. Studies whose authors were uncontactable or who did not reply to the initial email and the reminders within two-weeks were not included in the mediation analysis. However, because these papers reported univariate associations between both bullying and academic achievement and bullying and cognitive-motivational factors, they were included in the univariate meta-analysis.

### 2.5. Statistical Analysis

#### 2.5.1. Summary Measures

The extracted data was presented in a range of formats (e.g., correlations, odds ratios, log odds ratio, means and standard deviations). The adopted effect size format for the pooled effect size of each meta-analysis was Odds Ratio (OR) with 95% confidence intervals for each study, which was evaluated against the overall weighted effect size. A random effect model was used where the within and between study variability is taken into account, and thus is more generalisable than the fixed effects model [4,53].

For the studies that provided multiple effect sizes for the same variable (e.g., child-report and peer-report), the aggregated mean was calculated. This is to avoid duplication of results for the same samples. In addition, the weight, which is the inverse of variance, of each study will be shown in each meta-analysis. This will give the precision of each study [49].

#### 2.5.2. Heterogeneity and Moderators Analysis

The presence of significant heterogeneity indicates that variations in effect sizes is due to specific factors and moderators rather than errors in sampling (*Qb*). *I^2^* was used to measure the variability across studies [54] where values above 75% indicate that the variance between studies is due to moderators, while values below 25% are due to random error [55]. Moderator analyses were performed for categorical variables using ANOVAs for all moderators (design, assessment method for each variable, age, country, bullying types and subgroups) separately for each predictive model (bullying–motivational factors; bullying–academic achievement; and the mediation effect of motivational factors between bullying and academic achievement).

#### 2.5.3. Publication Bias

Four methods were used to calculate publication bias, each method giving a different indication. First, the Rosenthal’s Failsafe Number [56] will specify the number of further studies that need to be published in order to nullify the significant results. If the reported Failsafe N exceeds the outcome of the equation 10 (5k + 10) (k: number of reported studies) then the results are not biased [57]. Secondly, the Begg and Mazumbar Rank Correlation Test (Kendall’s tau b) [58] examines study sample size where small studies and large effect sizes indicate large variances. Thus, no publication bias means that the relationship between the effect size and variance is not significant. Like correlation, Kendall tau b with a value of zero indicates no correlation and the deviation from zero means that there is an association [59]. Thirdly, the Egger’s test [60] uses linear regression to calculate deviations from zero in the funnel plot. The higher the deviation from zero the larger the systematic difference between larger and smaller studies. Finally, Duval and Tweedie’s Trim and Fill Test [61] removes asymmetric studies from one side in order to identify the unbiased effect. These studies are then reinserted to create a symmetric funnel plot, and then an adjusted effect size is calculated for this symmetric plot [62]. The deviation between effects sizes will give an indication of the severity of publication bias. 

One study removal analysis was also performed for each meta-analysis to show whether any study’s removal would affect the significance level of the pooled effect size.

## 3. Results

### 3.1. Search Results

EndNote program [63] was used for importing studies from different databases. The first databases search produced 401 articles. Duplicates and articles that did not meet the inclusion criteria were removed firstly according to titles and abstracts and then according to full text reviews. All reference lists in the included articles and meta-analyses were also reviewed. The final number of articles that investigated the three main factors (bullying, one cognitive–motivational factor and academic achievement) and were included in the meta-analysis was 11 (Figure 1).

### 3.2. Study Characteristics

The 11 studies included 257,247 children and adolescents (number ranged between 140–235,064) aged between 5 and 17 years-old (Table 1). All studies included both genders, such that 52.07% of the participants included in the meta-analysis were female. Most studies only looked at bullying victimisation (N = 7), whereas two studies also reported findings on victims, bullies, and bully–victims, and one study reported findings on both bullies and victims. Additionally, most studies were cross-sectional (N = 6), and were conducted in North America (N = 8) and Europe (N = 3).

Among the studies included in the meta-analysis, only one study directly tested the indirect effect of bullying on academic achievement through motivation that also included data on academic engagement [46]. In addition, seven studies tested the indirect effect of bullying on academic achievement through academic engagement, academic self-concept, self-esteem, or psychological distress [45,46,48,50,64,65,66]. These variables reflect students’ general motivation levels and thus we decided to include them in this meta-analysis, referring to them as cognitive-motivational factors. One study [50] did not test the indirect effect of victimisation on academic achievement, but because the direct effect of bullying on academic engagement and the direct effect of engagement on academic achievement were provided, we used these coefficients to calculate the indirect effect of victimisation on academic achievement. Therefore, seven studies (eight mediation results) were included in the meta-analysis of the indirect effect of bullying on academic achievement through cognitive-motivational factors. There were an additional four studies [15,51,52,67] that did not test mediation nor provide data that allowed for the calculation of the indirect effect. However, because such studies looked at both the association between bullying and academic achievement and bullying and motivational factors, we decided to present their effect size findings in the meta-analysis.

The informant regarding bullying, academic achievement, and cognitive-motivational factors was also reported (children, peers, school-report or a mixture of respondents and methods). Most papers (N = 9) relied on children’s self-reports to assess bullying, whereas a limited number of studies used peer nomination (N = 1), and mixed informants (N = 1). In turn, most studies relied on teacher-report or school-report to assess academic achievement (N = 6), whereas the remaining studies used children’s self-reports (N = 5). For motivational factors (mediator), most studies relied on children’s self-reports (N = 5), whereas two relied on teacher-report and the rest of the studies did not report (N = 4).

### 3.3. Bullying and Victimisation as Predictors of Motivational Factors and Academic Achievement: Meta-Analysis

The studies that included motivational factors as mediators as well as academic achievement as an outcome were included in the analysis. Some studies looked at cognitive–motivational factors and academic achievement separately without calculating the mediation effects (N = 5), which were excluded from the analysis.

Firstly, we will present the analysis of the relationship between bullying involvement and cognitive-motivational factors. Secondly, the analysis of the relationship between bullying involvement and academic achievement will be presented and finally we will present the mediation effect of motivational factors on the relationship between bullying involvement and academic achievement. For all categories, a pooled effect size across studies of Odds Ratio (OR) was calculated.

#### 3.3.1. Motivational Factors

The combined effect size showed that children who are involved in any bullying behaviour were significantly less likely to be academically engaged (k = 4) (OR = 0.571, 95% CI (0.43, 0.77), *p* = 0.000), to have less motivation (k = 7) (OR = 0.82, 95% CI (0.69, 0.97), *p* = 0.021), and to have lower self-esteem (k = 1) (OR = 0.12, 95% CI (0.07, 0.20), *p* = 0.000). However, neither self-concept (k = 3) (OR = 0.74, 95% CI (0.53, 1.03), *p* = 0.072) nor self-efficacy (k = 2) (OR = 0.73, 95% CI (0.41, 1.27), *p* = 0.264) were significantly associated with bullying involvement. The heterogeneity assessments were significant for all except self-esteem (academic engagement: *Q*(3) = 14.39, *p* = 0.002; *I^2^* = 79.16%; motivation: and *Q*(6) = 119.05, *p* = 0.000; *I^2^* = 94.96%; self-concept: *Q*(2) = 21.78, *p* = 0.000; *I^2^* = 90.82%; self-efficacy: *Q*(1) = 5.64, *p* = 0.018; *I^2^* = 82.27%). The pooled effect size for the overall cognitive–motivational factors was significant (k = 17; OR = 0.67, 95% CI (0.59, 0.76), *p* = 0.000) with a significant heterogeneity between groups (*Q*(16) = 442.71, *p* = 0.000; *I^2^* = 96.39%) (See Figure 2).

As for victims only, they were also significantly less likely to be motivated (k = 4) (OR = 0.74, 95% CI (0.61, 0.89), *p* = 0.002). The pooled effect size for the overall cognitive-motivational factors for victims only was significant (k = 13) (OR = 0.63, 95% CI (0.55, 0.72), *p* = 0.000) with a significant heterogeneity between groups (*Q*(12) = 424.96, *p* = 0.000; *I^2^* = 97.18%). On the other hand, the results for bullies only were not significant in relation to motivation (k = 2) (OR = 1.03, 95% CI (0.84, 1.27), *p* = 0.762). Figure 2 shows other individual relationships for each bullying subgroup (Figure 2).

#### 3.3.2. Academic Achievement

The combined pooled effect size showed that children who are involved in bullying behaviour were significantly more likely to have low academic achievement (k = 14) (OR = 0.61, 95% CI (0.47, 0.79), *p* = 0.000). The heterogeneity assessment was also significant (*Q*(13) = 974.27, *p* < 0.000, *I^2^* = 98.66%).

The results for victims only also showed significant results where victims were more likely to have low academic achievement (k = 10) (OR = 0.62, 95% CI (0.47, 0.83), *p* = 0.001) with a significant heterogeneity between groups (*Q*(9) = 966.67, *p* = 0.000; *I^2^* = 99.07%). The results for bully/victims only and bullies only were not significant (bully/victims: k = 2, OR = 0.58, 95% CI (0.18, 1.89), *p* = 0.367); bullies: k = 2, OR = 0.55, 95% CI (0.26, 1.19), *p* = 0.128) (Figure 3).

### 3.4. Mediation Analysis between Victimisation and Academic Achievement Pooled Effect Size

From the above, only seven studies reported mediation results (one of which reported two mediations) or have enough data to calculate the mediation effect between victimisation and academic achievement. The studies that were included in the mediation analysis showed different mediation factors between victimisation and academic achievement, while three studies showed similar mediation factor (academic engagement).

There were significant mediation effects between victimisation and academic achievement for psychological distress and academic engagement combined as one mediation factor (k = 1) (OR = 0.69, 95% CI (0.50, 0.97), *p* = 0.031), self-concept (k = 1) (OR = 0.26, 95% CI (0.16, 0.43), *p* = 0.000), self-efficacy (k = 1) (OR = 0.65, 95% CI (0.61, 0.68), *p* = 0.000), motivation (k = 1) (OR = 0.87, 95% CI (0.82, 0.91), *p* = 0.000) and academic engagement (k = 3) (OR = 0.77, 95% CI (0.59, 0.99), *p* = 0.044). On the other hand, no significant mediation was found for self-esteem and self-efficacy combined as one mediation factor (k = 1) (OR = 0.87, 95% CI (0.54, 1.39), *p* = 0.546).

The overall pooled effect size for all motivational factors as mediators was significant (k = 8) (OR = 0.74, 95% CI (0.72, 0.77), *p* = 0.000) with a significant heterogeneity between groups (*Q*(7) = 79.30, *p* = 0.000; *I^2^* = 91.17%) (See Figure 4).

### 3.5. Moderator Analysis

As the *I^2^* variance for all analyses was above 75%, this indicates that the differences were due to moderators and not a random error. Meta-ANOVAs were conducted for moderation analyses for the categorical moderators: assessment method of bullying (child-report, peer nomination or mixed); assessment method of the outcomes or mediators (child report, teachers’ report, or schools’ report); country; age (childhood: 5–10; adolescence: 11–18 or mixed); bullying type; and design (longitudinal or cross-sectional).

#### 3.5.1. Moderator Analysis of Motivational Factors

The heterogeneity analysis was significant for some of the analyses. Heterogeneity assessment was conducted for children who were involved in bullying in relation to cognitive–motivational factors.

The heterogeneity assessment indicated that motivational factors were significantly moderated by country (*Qb* = 28.85, *p* = 0.000). This indicated that bullying involvement had a stronger effect on motivational factors in Canadian studies (k = 2, OR = 0.46, *p* = 0.000) followed by studies in Spain (k = 4, OR = 0.57, *p* = 0.000), then American studies (k = 8, OR = 0.75, *p* = 0.000) and finally Austrian studies (k = 3, OR = 0.98, *p* = 0.86).

Finally, the heterogeneity assessment indicated that motivational factors were also significantly moderated by the type of bullying (*Qb* = 24.66, *p* = 0.000). This indicated that relational bullying had a stronger effect on motivational factors (k = 2, OR = 0.52, *p* = 0.014) followed by cyberbullying involvement (k = 4, OR = 0.57, *p* = 0.000), then sexual bullying (k = 1, OR = 0.72, *p* = 0.000) and traditional bullying (k = 10, OR = 0.81, *p* = 0.000).

On the other hand, heterogeneity was not significant for design, age category and assessment method for the mediator and bullying. However, when looking specifically at design, it was found that longitudinal studies (k = 12, OR = 0.70, *p* = 0.000) and cross-sectional studies (k = 5, OR = 0.68, *p* = 0.001) were significant. For age, studies that included adolescents (k = 11, OR = 0.74, *p* = 0.000), children (k = 1, OR = 0.69, *p* = 0.047) and mixed (children and adolescents) (k = 5, OR = 0.64, *p* = 0.000) were significant. In addition, studies that included children’s reports (k = 16, OR = 0.70, *p* = 0.000) and mixed reports (k = 1, OR = 0.69, *p* = 0.047) for bullying data were significant. Finally, studies that included children’s reports (k = 15, OR = 0.73, *p* = 0.000) and teachers’ reports (k = 2, OR = 0.52, *p* = 0.014) for the data on mediator factors were significant.

#### 3.5.2. Moderator Analysis for Academic Achievement

The heterogeneity analysis was significant for some of the analyses. Heterogeneity assessment was conducted for children who were involved in bullying in relation to academic achievement. The heterogeneity assessment indicated that this relationship was significantly moderated by age categories (*Qb* = 7.30, *p* = 0.026). This indicates that bullying involvement had a stronger effect on academic achievement for studies amongst children (k = 1, OR = 0.58, *p* = 0.003) and adolescents (k = 12, OR = 0.58, *p* = 0.005) followed by a study that included both children and adolescents (k = 1, OR = 0.83, *p* = 0.000).

The heterogeneity assessment indicated that the relationship was also significantly moderated by the assessment method for bullying data (*Qb* = 12.62, *p* = 0.002). This indicated that bullying involvement had a stronger effect on academic achievement in studies that had peer nomination (k = 3, OR = 0.23, *p* = 0.000) followed by a study where bullying was reported by a mix of informants (k = 1, OR = 0.56, *p* = 0.003) and finally by children’s reports (k = 10, OR = 0.71, *p* = 0.013).

The heterogeneity assessment indicated that the relationship was also significantly moderated by country (*Qb* = 21.41, *p* = 0.000). This indicated that bullying involvement had a stronger effect on academic achievement in studies in Portugal (k = 3, OR = 0.23, *p* = 0.000), followed by American studies (k = 6, OR = 0.64, *p* = 0.020), then Canadian studies (k = 2, OR = 0.66, *p* = 0.000) and finally Austrian studies (k = 3, OR = 0.85, *p* = 0.038).

The heterogeneity assessment indicated that the relationship was also significantly moderated by study design (*Qb* = 11.26, *p* = 0.001). This indicated that bullying involvement had a stronger effect on academic achievement in longitudinal studies (k = 7, OR = 0.43, *p* = 0.000) compared with cross-sectional studies (k = 7, OR = 0.85, *p* = 0.006).

Finally, the heterogeneity assessment indicated that the relationship was also significantly moderated by the type of bullying (*Qb* = 26.10, *p* = 0.000). This indicated that studies that investigated general bullying had a stronger effect on academic achievement (k = 3, OR = 0.23, *p* = 0.000), followed by studies that reported relational bullying involvement (k = 2, OR = 0.66, *p* = 0.000), then traditional bullying (k = 8, OR = 0.70, *p* = 0.119) and finally sexual bullying (k = 1, OR = 0.83, *p* = 0.000).

On the other hand, when looking specifically at assessment methods for the outcome (academic achievement), it was found that schools’ reports (k = 6, OR = 0.51, *p* = 0.005) and teachers’ reports (k = 1, OR = 0.69, *p* = 0.001) were significant, but not children’s reports (k = 7, OR = 0.67, *p* = 0.059).

#### 3.5.3. Moderator Analysis for Mediation

The heterogeneity analysis was significant for some of the analyses. Heterogeneity assessment was conducted for children who were involved in victimisation. The heterogeneity assessment indicated that the mediation analysis was significantly moderated by the assessment of the outcome (academic achievement) (*Qb* = 5.31, *p* = 0.070). This indicated that mediational factors had a stronger effect on academic achievement for teachers’ reports (k = 2, OR = 0.83, *p* = 0.267), compared to children’s reports (k = 6, OR = 0.66, *p* = 0.000).

On the other hand, when looking specifically at design, it was found that longitudinal mediation studies were significant (k = 5, OR = 0.76, *p* = 0.004), but not cross-sectional mediation studies (k = 3, OR = 0.55, *p* = 0.069). For age, mediation studies that included adolescents (k = 7, OR = 0.70, *p* = 0.000) and children (k = 1, OR = 0.67, *p* = 0.029) were significant. In addition, studies that included children’s reports (k = 7, OR = 0.70, *p* = 0.000) and mixed reports (k = 1, OR = 0.67, *p* = 0.029) for the bullying data were significant, while studies that included children’s reports (k = 4, OR = 0.72, *p* = 0.002) on academic achievement were significant, but not schools’ reports (k = 3, OR = 0.55, *p* = 0.069) or teachers’ reports (k = 1, OR = 0.96, *p* = 0.658). Mediation data studies that included children’s reports (k = 6, OR = 0.66, *p* = 0.000) were significant while teachers’ reports were not significant (k = 2, OR = 0.83, *p* = 0.267). Studies that reported traditional bullying were significant (k = 6, OR = 0.66, *p* = 0.000) but not studies on relational bullying (k = 2, OR = 0.83, *p* = 0.274).

### 3.6. Publication and Risk Bias

Four publication bias methods were employed (see Table 2). The studies included in each analysis are reflected in each meta-analysis figure shown above. For example, when the motivational factors are the outcome and the predictor is any bullying involvement then the analysis was done for all studies shown in Figure 2, while when the predictor is the victims only subgroup then the studies that investigated victimisation only are included, and so on.

#### 3.6.1. Cognitive-Motivational Factors as Outcomes of Bullying Involvement

The ‘5k + 10’ benchmark using the Rosenthal’s Failsafe N analysis was not reached for bullies only, indicating that the found effects are open for future disconfirmation. The rest Rosenthal’s Failsafe N analyses indicated no publication bias for any bullying involvement and victimization only. The Kendall’s Tau calculations and Egger’s Test for all indicated an absence of publication bias. Lastly, the Trim-and-Fill analysis did not show different effect sizes for any of the results.

#### 3.6.2. Academic Achievement as an Outcome of Bullying Involvement

For bullying involvement and victimisation there was no publication bias as the ‘5k + 10’ benchmark was reached. The Kendall’s Tau calculations indicated publication bias for any bullying involvement, but not for victims only. The Egger’s Test showed no publication bias. Lastly, the Trim-and-Fill analysis showed exactly the same effect sizes for both.

#### 3.6.3. Mediation

For the mediation studies there was no publication bias as the ‘5k + 10’ benchmark was reached. The Kendall’s Tau calculation and the Egger’s Test did not find any publication bias. The Trim-and-Fill procedure showed exactly the same effect size for the mediation.

In addition, the academic engagement studies showed publication bias, as the ‘5k + 10’ benchmark was not reached. No publication bias was found in the rest of the tests (See Table 2).

#### 3.6.4. One Study Removed

We repeated the meta-analyses by removing each study one by one. The results show that when removed none of the studies changed the pooled effect sizes for the relationship between bullying and/or victimisation involvement and academic achievement (See Figure 5), for the relationship between bullying and/or victimisation involvement and cognitive motivational factors (See Figure 6), and for the mediation analysis (See Figure 7). The pooled effect sizes remained significant as shown in the original analysis before removing any of the studies, indicating that none of the studies could change the results when removed.

## 4. Discussion

The purpose of this meta-analysis was to investigate the effect of cognitive–motivational factors on the association between bullying involvement and academic achievement. To our knowledge this is the first meta-analysis that investigated the mediation between bullying involvement and academic achievement. The overall findings showed that the relationship between victimisation and academic achievement was significantly mediated by cognitive–motivational factors. Specifically, victimisation was associated with low scores on the cognitive–motivational factors evaluated (e.g., motivation, student engagement), which were, in turn, associated with low academic achievement. These relationships were moderated by country, where American studies from the US had stronger mediation than one Canadian study. In addition, longitudinal studies and studies that included traditional victimisation had a significant mediation effect while cross-sectional studies and studies on relational victimisation did not.

The overall finding that bullying involvement and specifically victimisation is associated with low motivation, which is linked to low academic achievement, is supported partially by the literature [8,36,37]. The previous literature looked at the relationship between bullying involvement and academic achievement separately without taking into account the cognitive-motivational factors. The cognitive-motivational factors in this study included motivation, academic engagement, self-esteem, self-efficacy and self-concept, which have been found in the literature to be affected by victimisation.

The question is: why do victims perform more poorly in their academic achievement compared to other children who are not bullied? What are the mechanisms that are behind this? This meta-analysis study points out to an important mediation of this relationship, namely cognitive–motivational factors. These mechanisms are supported by some theories. These include the self-determination theory (SDT) [38], the expectancy–value theory [32,33,68], and the achievement goal theory [34,35], that may explain these relationships. Firstly, those who are bullied are more likely to be less motivated [21] and have lower aspiration [31] to engage in a particular goal such as academic success and achievement. Secondly, victims may have a negative view of themselves [19], have low self-efficacy (the belief in one’s ability to succeed in specific situations or accomplish a task) [23,69] and have low self-concept [24] that can in turn affect their scholastic achievement [36]. Thirdly, victimisation may lead to isolation, school adjustment problems including loneliness, and school avoidance [41] and as a result their self-esteem and their self-efficacy are also affected. These in turn put these children at risk of school absenteeism [70], truancy (e.g., [71]), and dropping out of school [72] as they may view their school as an unsafe place (e.g., [73]). For example, Jan and Husain [74] found that bullied students were more likely to miss school for fear of being criticized by their peers and Buhs et al. [75] found that chronically abused children were more likely to engage in school avoidance behaviour. Fourth, peer victimisation may also result in internalizing problems [41,76] and somatic and psychological problems [41,77] that result in problematic levels of school absenteeism [73,78,79], which, in turn, results in poor academic outcomes [80]. This could also lead to less engagement as they are afraid of being mocked and made fun of and as a result perform more poorly. In a longitudinal study, Juvonen et al. [81] found that peer harassment led to psychological maladjustment (low self-worth, loneliness, and depressive symptoms), which led to poor school functioning. Children with depressive symptoms may exhibit poor concentration and memory, and consequently, have low academic achievement [82]. Finally, victims may also have a negative perception of their school climate [13,73], which may, in turn, cause school absenteeism [83], and poor academic outcomes [9,84,85].

### 4.1. Implications

The finding that victimisation affects both motivational factors and academic achievement has great implications for educational practice. Educational interventions that aim to improve academic success and achievement need to take into account these aspects. The first step of an intervention program should therefore be decreasing victimisation [86,87] and particularly focusing on improving motivation, self-esteem, self-efficacy and self-concept. The second step is to increase students’ academic achievement and enhance their educational engagement.

In addition, high levels of support from family and friends [42,88,89] and a positive teacher–child relationship that can have a positive effect by impacting their sense of school connectedness [90] can protect bullied children and adolescents from poor academic outcomes.

Studies that looked at the mediation relationship are very few and those few studies looked at different motivational factors as discussed above. Given that the findings of the meta-analysis showed that there is an indirect negative effect of victimisation on academic success and achievement through cognitive–motivational factors, this is an area of research yet to be explored further with these factors and to include all bullying subgroups (bullies, victims, and bully/victims) and types (direct, relational, and cyber). There is a particular need for longitudinal studies examining whether bullying in fact precedes changes in the cognitive–motivational factors examined in the current study that in turn impair academic achievement. Despite the increasing number of studies on bullying and victimisation, most of the mediation studies were based in only a few countries. Therefore, there is also a need for studies in multiple countries, particularly in developing countries. This is important as different countries and cultures deal and define bullying and victimisation differently [91,92] and have different educational, school and grading systems, and thus interventions may differ in each country accordingly.

### 4.2. Strengths and Limitations

This is the first meta-analysis study that looked at the mediation effect of cognitive-motivational factors on the relationship between bullying victimisation and academic achievement. The study pointed out the lack of studies in this area and the need for more studies on these mediation factors. The study also gave a good insight into the mechanism for why victimisation can lead to lower academic achievement. This can inform policy makers, practitioners (psychologists, educationalists) and future interventions of the best way to improve victims’ school achievement by concentrating on these factors.

The process of coding and grouping of related terms is a key factor in a meta-analysis. Studies usually differ slightly in terminology and methodologies [91,92]; nevertheless, groups need to be formed, in order to study them in a meta-analysis. Vast methodological differences between cognitive–motivational factors were also observed. However, the study indicated that these factors negatively mediated the relationship between victimisation and academic achievement. This should be further investigated with more studies on these factors. Similarly, the studies usually included victimisation without looking at bullying others and bully/victims.

The term ‘motivation’ is quite broad and the grouping of related yet distinctive terms (e.g., self-esteem, self-efficacy, self-concept, motivation and academic engagement) might have overshadowed underlying types of motivation. The multifaceted dynamics of motivation need to be explored in greater detail. However, the findings of the current study are an important platform for this type of investigation.

This study utilised four methods to investigate possible publication biases. There were some publication biases especially in relation to bullies and bully/victims simply due to the small number of studies, while for some cases, publication bias was not performed as there were less than three studies (bullies only and bully/victims) and thus these areas should be further investigated. However, our results can be perceived as relatively robust especially with regards to victimisation. In addition, one study removal did not affect the final pooled effect sizes and all results remained significant.

## 5. Conclusions

The current study is the first meta-analysis that examined the mediation effect of different cognitive–motivational factors on the relationship between bullying victimisation and academic achievement, including moderators. These results showed that motivational factors negatively mediated these relationships. Additionally, the effect sizes were moderated by some moderators including the design of the studies, age, assessment methods for reporting bullying, mediators and outcomes, countries and/or bullying types. Only few studies as shown here looked at the mediation effect of motivation, while none of the studies included aspiration as a mediator. In addition, these studies looked at different cognitive–motivational factors, which shows the need for more studies in this area.

The findings of this meta-analysis are important for educational and psychological practitioners, parents and schools [79,93]. Based on these findings intervention programs and anti-bullying policies [94,95] need to be implemented in schools and parents and family dynamics should play a central role in these interventions. In addition, interventions can concentrate on internal and external motivational and academic factors. Motivational factors can serve as protective factors in these situations, therefore positive enforcement, encouragement, and programs for engaging these children and adolescents should be designed. Furthermore, the findings highlight the need for further studies on each cognitive-motivational factor including several moderators.

## Figures and Tables

**Figure 1 ijerph-18-02209-f001:**
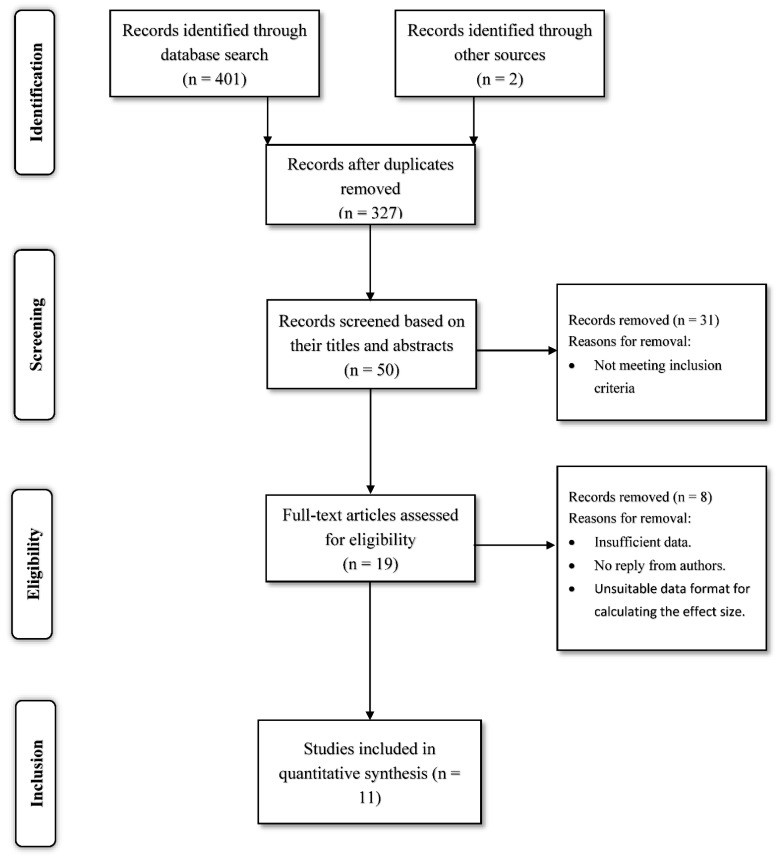
Description of the systematic search stages.

**Figure 2 ijerph-18-02209-f002:**
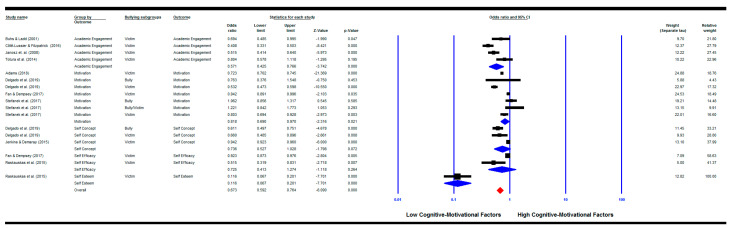
Meta-analysis for the relationship between bullying involvement and cognitive-motivational factors.

**Figure 3 ijerph-18-02209-f003:**
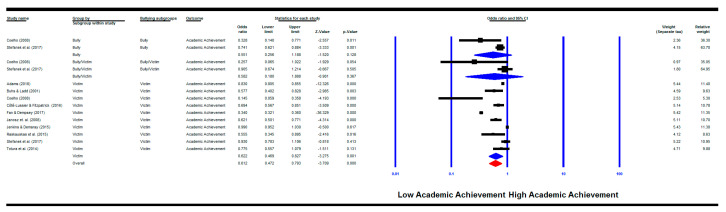
Meta-analysis of the relationship between bullying involvement and academic achievement.

**Figure 4 ijerph-18-02209-f004:**
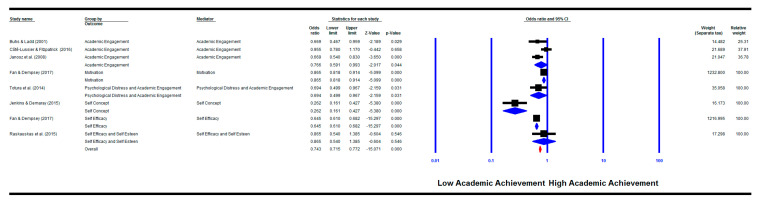
Meta-analysis showing the mediation effect of cognitive–motivational factors between victimisation involvement and academic achievement.

**Figure 5 ijerph-18-02209-f005:**
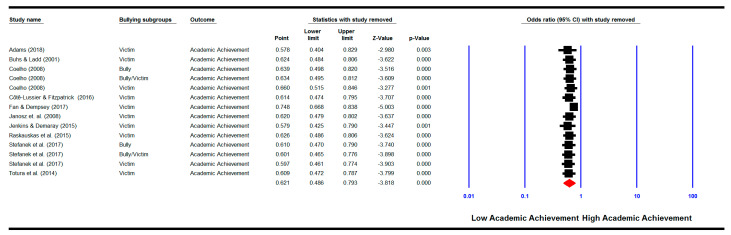
‘One study removed’ analysis: meta-analysis showing the pooled effect size of the relationship between bullying involvement and academic achievement with each study removed.

**Figure 6 ijerph-18-02209-f006:**
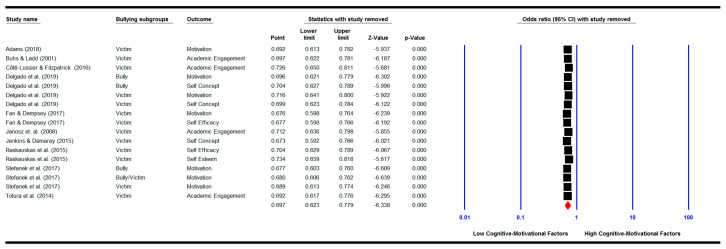
‘One study removed’ analysis: meta-analysis showing the pooled effect size of the relationship between bullying involvement and cognitive-motivational factors with each study removed.

**Figure 7 ijerph-18-02209-f007:**
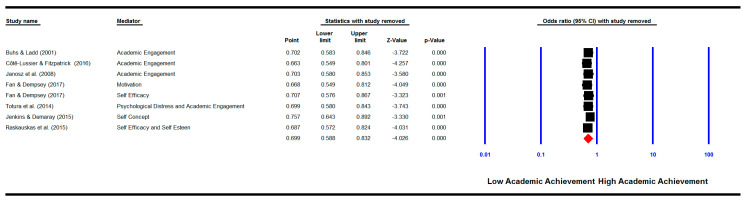
‘One study removed’ analysis: meta-analysis showing the pooled effect size of the mediation effect of cognitive-motivational factors on the relationship between victimisation involvement and academic achievement with each study removed.

**Table 1 ijerph-18-02209-t001:** Study characteristics for the studies included in the meta-analysis.

Study	N	% of Females	Age Range	Country	Design *	Type of Bullying/Victimization †	Bullying Subgroup	Type of Assessment (bullying) ‡	Type of Assessment (Mediator)	Type of Assessment (Outcome: Academic Achievement)
Adams (2018)	235,064	-	10–18	USA	Cross-sectional	Sexual orientation victimization	Victims	Child self-report	-	Child self-report
Buhs & Ladd (2001)	399	49.40	5.52	USA	Longitudinal (Between Fall and the Spring)	Relational bullying	Victims	Mixed (Teacher’s, student’s, and peer’s report)	Teacher-report	Child self-report
Coelho (2008)	355	53.70	12–17	Portugal	Longitudinal (Six months)	General bullying	Victims, bullies, bully-victims	Peer nomination	-	School-report
Cotê-Lussier & Fitzpatrick (2016)	1234	54	13	Canada	Longitudinal (One year)	Relational bullying	Victims	Child self-report	Teacher-report	Teacher-report
Delgado et al. (2019)	548	49.80	10–13	Spain	Cross-sectional	Cyberbullying	Victims, bullies	Child self-report	-	Child self-report
Fan & Dempsey (2017)	16,252	50.10	15–16	USA	Longitudinal (Four years)	Traditional bullying	Victims	Child self-report	Child self-report	School-report
Janosz et al (2008)	1104	48.40	11–15	Canada	Longitudinal (One year)	Traditional bullying	Victims	Child self-report	Child self-report	Child self-report
Jenkins & Demaray (2015)	140	58.60	11–14	USA	Cross-sectional	Traditional bullying	Victims	Child self-report	Child self-report	Teacher-report
Raskauskas, Rubiano, Offen, & Wayland (2015)	231	55.90	12–15	USA	Cross-sectional	Traditional bullying	Victims	Child self-report	Child self-report	School-report
Stefanek, Strohmeier, & Yanagida (2017)	1451	48.80	12–15	Austria	Cross-sectional	Traditional bullying	Victims, bullies, bully-victims	Child self-report	-	Child self-report
Totura, Karver, & Gesten (2014)	469	53.60	11–14	USA	Cross-sectional	Traditional bullying	Victims	Child self-report	Child self-report	School-report

* For longitudinal studies the duration of the study is mentioned between brackets. † Definitions: traditional bullying included face-to-face bullying including relational bullying (i.e., purposeful damage and manipulation of peer relationships leading to social exclusion, spreading rumours) and direct bullying (i.e., physical such as hitting, pushing, and verbal such as making fun, insulting someone); cyberbullying included bullying through digital electronic communication; general bullying included violence and intimidation based on peer nomination of up to three students in the class that are perpetrators based on three items (starts fights, says unpleasant things, and gets upset easily) and victims (gets teased, gets picked on, gets pushed or hit). ‡ The predictor (bullying) and the mediator variables were measured through standardised measures in all studies. Academic achievement was measured either through the students’ self-report GPA, or through schools’ or teachers’ reports.

**Table 2 ijerph-18-02209-t002:** Publication bias analysis using four methods.

Outcome	Predictor	Publication Bias Methods				
Motivational factors		Fail Safe Na	‘5k + 10’ Benchmark	Begg and Mazumdar (Kendall’s Tau)	Egger’s Test (95% CI)	Trim-and-Fill (95% CI)
	Any bullying involvement	1666	95	−0.088 *p* = 0.621	β = −2.86 (−6.05, 0.33) *p* = 0.07	0.69 (0.62, 0.78)
	Victims only	1513	75	−0.21 *p* = 0.328	β = −4.06 (−8.26, 0.13) *p* = 0.056	0.66 (0.58, 0.75)
	Bullies only	4	37	0.33 *p* = 0.601	β = 0.07 (−67.03, 67.17) *p* = 0.991	0.80 (0.51, 1.24)
	Bully/Victims ^1^	NA	NA	NA	NA	NA
Academic Achievement	Any bullying involvement	1546	80	−0.45 *p* = 0.025	β = −2.39 (−8.99, 4.21) *p* = 0.444	0.62 (0.49, 0.79)
	Victims only	1226	60	−0.42 *p* = 0.089	β = −3.24 (−13.88, 7.39) *p* = 0.501	0.62 (0.47, 0.83)
	Bullies only ^2^	NA	NA	NA	NA	NA
	Bully/Victims ^2^	NA	NA	NA	NA	NA
Mediation analysis	Victimisation (for all studies)	308	50	−0.21 *p* = 0.457	β = −0.83 (−5.34, 3.69) *p* = 0.669	0.70 (0.59, 0.83)
	Victimisation (for Academic engagement studies, N:3)	8	25	−0.33 *p* = 0.601	β = −3.10 (−75.86, 69.66) *p* = 0.684	0.79 (0.59–0.99)
Motivational factors	Any bullying involvement	1666	95	−0.088 *p* = 0.621	β = −2.86 (−6.05, 0.33) *p* = 0.07	0.69 (0.62, 0.78)
	Victims only	1513	75	−0.21 *p* = 0.328	β = −4.06 (−8.26, 0.13) *p* = 0.056	0.66 (0.58, 0.75)
	Bullies only	4	37	0.33 *p* = 0.601	β = 0.07 (−67.03, 67.17) *p* = 0.991	0.80 (0.51, 1.24)
	Bully/Victims ^1^	NA	NA	NA	NA	NA
Academic Achievement	Any bullying involvement	1546	80	−0.45 *p* = 0.025	β = −2.39 (−8.99, 4.21) *p* = 0.444	0.62 (0.49, 0.79)
	Victims only	1226	60	−0.42 *p* = 0.089	β = −3.24 (−13.88, 7.39) *p* = 0.501	0.62 (0.47, 0.83)
	Bullies only ^2^	NA	NA	NA	NA	NA
	Bully/Victims ^2^	NA	NA	NA	NA	NA
Mediation analysis	Victimisation (for all studies)	308	50	−0.21 *p* = 0.457	β = −0.83 (−5.34, 3.69) *p* = 0.669	0.70 (0.59, 0.83)
	Victimisation (for Academic engagement studies, N:3)	8	25	−0.33 *p* = 0.601	β = −3.10 (−75.86, 69.66) *p* = 0.684	0.79 (0.59, 0.99)

^1^ Only one study and thus cannot be performed; ^2^ Only two studies and thus cannot be performed.

## Data Availability

Data is contained within the article and Appendix A.

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
