# Peer review of "How Can Bullying Victimisation Lead to Lower Academic Achievement? A Systematic Review and Meta-Analysis of the Mediating Role of Cognitive-Motivational Factors"

_ijerph, 2021, doi:10.3390/ijerph18052209_

Round 1
Reviewer 1 Report
Review of the manuscript: How bullying victimization lead to lower academic achievement? A systematic review and meta-analysis of the mediating role of cognitive-motivational factors.
Manuscript ID: 1071667
General Comment Points
This manuscript reports on an interesting meta-analytic review on the mediation effect of motivation and aspiration on the relationship between peer victimization and academic achievement. I applaud the aspirations represented in this paper. However, both formal and content aspects of the manuscript must be revised. I hope the suggestions I give below will support you in advancing your research efforts on this topic. Following are my specific comments on this paper.
Title Review Points
The title captures the reader’s attention and clearly informs the reader about the contents of the article. I have no suggestions for improving the title.
Abstract Review Points
- The authors should included previous review and meta-analysis and the gap.
- More information on the methodology could be included in the abstract.
- The authors should remove statistical data from the abstract
- They should describe more the participants. For example, age, sex, and ethnic.
- Moreover, they should include the essential features of study method.
- The authors should improve the implications and applications.
Introduction Review Points
- There is a clear statement of the purpose in the introduction. However, the information is scarce. The introduction does not summarize, integrate, and critically evaluate the reviews and meta-analysis on the topic of this paper.
- There are no hypotheses.
- The authors should include the contributions of the study.
Method Review Points
- The authors should improve data analysis description.
Discussion Review Points
- The discussion should be divided into subsections
- There is not contributions of this study and justify why the findings are important
Reviewer 2 Report
Strengths:
- Nice description why this is a relevant topic and what the theory behind the research questions is.
- The Research Questions are clearly described.
- Good Method section and PRISMA flow diagram.
- The description of how the data were analyzed is clarity.
- Conclusions supported by data.
Limitations
-The discussion chapter needs to be reorganized and rewritten.
In my opinion, the study is very interesting and worth being published. In fact, the article is well-structured, the method used is congruent with the purpose of the stud and the results are presented clearly. Nevertheless, I feel I can give the following suggestions:
I am afraid that the discussion chapter still needs work. I here offer some insight into how the discussion part should be re-written.
- First of all, I suggest to include one short paragraph summarizing the purpose of the study.
- Second, I think that the whole discussion needs to be re-structured to include: theoretical implications of the study, practical implications and a brief conclusion.
I think the authors can easily follow the suggestions I have given in this review and make a new version of their interesting paper.
All best wishes.
Reviewer 3 Report
The article makes an important contribution to the understanding of the interface between bullying victimization and lower academic achievement, examining the mediation effect of cognitive-motivational factors such as academic motivation and aspiration.
The article is written in clear and correct language. However, I will highlight some points that need to be clarified by the authors.
ABSTRACT
It is well structured and written in a concise and easy-to-read form, highlighting the main points of the article.
INTRODUCTION
It presents the study problem in a concise and well-structured manner, placing the significance of the study on the basis of relevant and updated literature and defining the objectives of the study.
Page 2 – Line 59: What is GPA? I suggest putting the term in full before putting the abbreviation of the terms
METHODS
- Why the protocol for this study was not registered in the PROSPERO records base (International Prospective Register of Systematic Reviews)?
- I suggest putting the term in full before putting the abbreviation of the terms: CINAHL and ERIC. Although they are known bases, authors must comply with writing rules.
- The search strategy needs to be more detailed. The search term combinations are unclear.
- The types of bullying need to be defined. What the authors considered traditional, relational, direct, and sexual bullying?
- How did the included articles assess bullying, academic achievement and cognitive-motivational factors? Did they use standardized instruments? This information is not described either in the methods or in the results.
- Page 4 – Topic 2.4
The authors cite that they converted the school grade into an average age for 3 studies that put the school grade instead of the children's age. In my opinion, these 3 studies could not be analyzed, because if the children were behind in school, the school grade could not be a proxy for age.
- The statistical analysis needs to be more detailed.
- Was a subgroup analysis performed by study country, study design, sex, age?
- Why did the authors evaluate only the publication bias? Why didn't they also evaluate the quality of the evidence by Grading of Recommendations Assessment, Development, and Evaluation (GRADE) system (for example)?
RESULTS
In the entire description of the results, I suggest that the references of the included articles be placed. As it is placed in the text, the reader does not know which studies presented that result. For example, in the sentences: “One study did not test the indirect effect of victimisation on academic achievement, but because the direct effect of bullying on academic engagement and the direct effect of engagement on academic achievement were provided, we used these coefficients to calculate the indirect effect of victimisation on academic achievement. Therefore, seven studies (eight mediation results) were included in the meta-analysis of the indirect effect of bullying on academic achievement through cognitive-motivational factors. There were an additional four studies that did not test mediation nor provide data that allowed for the calculation of the indirect effect. However, because such studies looked at both the association between bullying and academic achievement, and bullying and motivational factors, we decided to present their effect size findings in the meta-analysis.”
Table 1:
I suggest putting more details of the studies.
For example:
- What was the study period, mainly for longitudinal studies? Only the year of the study was cited.
- What are the definitions of bullying in each study? Is general bullying the same or different from traditional bullying?
- How did children, peers, teachers assess bullying, mediators and outcome?
Page 8 – Lines 338 e 339: Put the % in the value of I2.
Figure 2 and 3: Where type of bullying is placed is bullying role.
Table 2: Which studies were included in this analysis?
DISCUSSION
It is well structured, dialogues the results with the previously published literature. It presents the potential limitations of the study, but it presents implications and consistent conclusions to facilitate future research and clinical practice.
REFERENCES
The reference 70 is incomplete.
Round 2
Reviewer 3 Report
After the adjustments have been made, the manuscript is consistent for publication.